# High Positive MR and Energy Band Structure of RuSb_2+_

**DOI:** 10.3390/ma13143159

**Published:** 2020-07-15

**Authors:** Liang Zhang, Yun Wang, Hong Chang

**Affiliations:** School of Physical Science and Technology, Inner Mongolia University, Hohhot 010021, China; zl1184078974@aliyun.com (L.Z.); wangyunqw@outlook.com (Y.W.)

**Keywords:** RuSb_2_, weak localization, magnetoresistivity

## Abstract

A high positive magnetoresistance (MR), 78%, is observed at 2 K on the *ab* plane of the diamagnetic RuSb_2+_ semiconductor. On the *ac* plane, MR is 44% at 2 K, and about 7% at 300 K. MR at different temperatures do not follow the Kohler’s rule. It suggests that the multiband effect plays a role on the carrier transportation. RuSb_2+_ is a semiconductor with both positive and negative carriers. The quantum interference effect with the weak localization correction lies behind the high positive MR at low temperature. Judged from the ultraviolet–visible spectra, it has a direct band gap of 1.29 eV. The valence band is 0.39 eV below the Fermi energy. The schematic energy band structure is proposed based on experimental results.

## 1. Introduction

Magnetoresistance (MR) is applied in many fields as magnetic memories, magnetic valves, magnetic sensors, and magnetic switches. In spin-polarized materials, MR is observed as the spin-up and spin-down carriers experience different conduction routes. Semiconductors are an important series of materials with many interesting properties [1,2]. MR is also observed in heavily doped n-type nonmagnetic semiconductors, such as Ge, Si, GaAs, and CdS [3,4,5,6]. In the last decade, MR in organic semiconductors has been one of the hottest research fields [7,8,9,10]. In organic semiconductors, the magnetic field generates secondary charge carriers due to dissociation and charge reaction [7,8,9,10]. Space charges accumulated at the organic–electrode interfaces change the injection current and account for the tunable MR. Recently, an extremely large MR is observed in Dirac semimetals Cd_3_As_2_ [11], Weyl semimetals (Mo, W)Te_2_ [12,13], and Bi_2_Te_3_ [14,15]. It is proposed that MR in semimetals is due to the increasing severity of the stringency of the hole/electron = 1 resonance with increasing magnetic field [12,13].

Recently, AB_2_-type compounds, with A as heavy transition metals and B as Te, Se, Sb, etc., have attracted a lot of attention due to a series of fascinating properties, such as topological insulating, superconducting, and huge MR [12,13,14]. In the past, RuSb_2_ has attracted research attentions with its thermoelectric properties. Different from other thermoelectric compounds, in which the Seebeck coefficient decreases with temperature, RuSb_2_ has a Seebeck coefficient peak at about 10 K. It is assumed that the huge Seebeck coefficient peak at low temperature is due to its unique band structure [16,17]. In theoretical analysis, the energy band gap is formed by the separation between the lifted *d_xy_* orbital of the Ru/Fe atom and the rest *t_2g_* doublets [18]. Ru_1−*x*_Mn*_x_*Sb_2+d_ single crystal has also shown thermoelectric properties [19]. Even though high MR is observed in devices combining a semiconductor and a magnetic material, such as in spin valve transistors [20], magnetic tunnel transistors [21], and stray-field-induced MR devices [22], MR is seldom observed in nonmagnetic single crystals. In nonmagnetic single crystals, MR is closely related to the changed electronic density of state by the applied magnetic field. It is able to act as an indicator of some other interesting physical phenomena. In this paper, we studied a high positive MR of RuSb_2+_ single crystal. Since no experimental work has ever been done on the energy band structure of RuSb_2_, we also proposed an energy band structure based on the information deduced from the electronic transportation, UV–vis spectra, and XPS valence band.

## 2. Experiments

RuSb_2+_ single crystal was grown with the self-flux method in a ratio of Ru:Sb = 1:10. High-purity Ru and Sb powder (99.8%, Alfa Aesar, Haverhill, MA, USA) from Alfa Aesa were mixed together and placed in an evacuated quartz tube. The samples were heated up to 1150 °C at a rate of 150 °C/h and kept at 1150 °C for 36 h. The samples were cooled down to 700 °C at a rate of 2 °C/h, and after that, the extra Sb flux was decanted in a centrifuge. In our previous paper, RuSb_2+_ with an extra Sb was proved by the energy dispersion X-ray spectra (EDX) on a Hitachi S-4500II field-emission scanning electronic microscope (Hitachi, Tokyo, Japan). The phase of the as-grown crystals was characterized using a powder X-ray diffractometer (Bruker D8 Advance, Bruker, Billerica, MA, USA) using Cu K_a_ radiation. Single-crystal X-ray diffraction (XRD) was carried out using Bruker Apex II X-ray diffractometer (Bruker Apex II, Billerica, MA, USA) with Mo radiation K_a1_ (λ = 0.71073 Å). Electrical transport and Hall effect between 2 K and 400 K were measured on a quantum design physical property measurement system (PPMS, Quantum Design, San Diego, CA, USA). The resistivities were measured using the standard 4-probe technique.The ultraviolet–visible (UV–vis) spectra were obtained using a PerkinElmer Lambda750 UV–vis spectra (PerkinElmer Lambda750, PerkinElmer, Kumamoto, Japan) at room temperature in the wavelength from 200 to 1500 nm with the sampling pitch of 2 nm. X-ray photoelectron spectroscopy (XPS) was carried out on a Thermo ESCALAB 250Xi with Al Ka photons (hv
*=* 1486.6 eV) and a hemispherical energy analyzer (Thermo ESCALAB 250Xi, Thermo, Waltham, MA, USA).

## 3. Results and Discussions

The typical size of the shiny RuSb_2+_ crystal is about 4 × 4 × 4 mm^3^, as shown in Figure 1a. EDX measurement, as shown in Figure 1b, gives the ratio of Ru:Sb = 29.9:70.1 = 1:2.3. It is close to the composition of RuSb_2_. The single-crystal XRD pattern, as shown in Figure 1c, is the procession image of the (h k 0) plane of RuSb_2+_ with the space group *Pnnm* [16]. In order to confirm the structure, the XRD patterns are measured on the *ab* and *ac* planes of the RuSb_2+_ single crystal, as shown in Figure 1d. The peaks in the pattern are exactly from RuSb_2+_ with the space group *Pnnm*. The lattice constants are *a* = 0.5951 (2) nm, *b* = 0.6674 (1) nm, and *c* = 0.3179 (1) nm.

Figure 2a shows the temperature dependence of the resistivity measured on the *a**b* and *ac* planes with the magnetic field out of the plane, $\rho_{ab}$ and $\rho_{ac}$, with crystal size of 1.5 × 1.5 × 1.7 mm^3^. Below the temperature of about 12 K at 0 T, which is named as *T*_f_, $\rho_{ab}$ has a very slow upturn. In fact, it is almost flat at 0 T. At 7 T, the slow upturning resistivity remains. At 14 T, $\rho_{ab}$ sharply increases at low temperature. A similar robust flat resistivity is also observed in some semimetals [11,12]. In a semimetal, the bottom of the conduction band narrowly overlaps with the top of the valence band. At applied magnetic fields, the carrier concentration changes at the Fermi surface. As a result, MR appears. As the temperature is above *T*_f_, $\rho_{ab}$ decreases.

In a single crystal, the grain boundary contribution is excluded. As the flat (or slow upturning at magnetic field) resistivity at low temperature is related to the semimetal, the quantum interference effects with the weak localization correction are used to explain MR [23]. The total resistivity in the first order is given by
(1)$$\rho (H,T)=\rho_{0}-\rho_{0}^{2}[\sigma_{ee}(H,T)+\sigma_{wl}(H,T)]$$
with $\rho_{0}$ as the residual resistivity, *H* as the magnetic field, $\sigma_{ee}$ as the conductivity caused by electron–electron interaction effects, and $\sigma_{wl}$ by weak localization, respectively. $\sigma_{ee}$ is expressed as
(2)$$\sigma_{ee}=\frac{e^{2}}{4\pi^{2}\hbar }1.3(\frac{4}{3}-\frac{3}{2}F)\sqrt{\frac{k_{B}T}{2\hbar D}}-\frac{e^{2}}{4\pi^{2}\hbar }F\sqrt{\frac{k_{B}T}{2\hbar D}}g_{3}(h)$$
with $h=g\mu_{B}\mu_{0}H/(k_{B}T)$, the Planck constant ℏ, the diffusion constant *D*, and the interaction constant *F*. $g_{3}(h)$ is a function of h and can be calculated numerically [23]. While the weak localization effect is suppressed by the high magnetic field, the electron–electron interaction is hardly affected [23]. Therefore, the resistivity at 14 T can be interpreted as the electron–electron interaction. In Equation (2), $\sigma_{ee}$ is proportional to $\sqrt{T}$. As shown in Figure 2b, below 25 K, the conductivity at 14 T is linearly fitted with $\sqrt{T}$. It suggests that electron–electron interaction affects the resistivity.

The weak localization contribution at zero field is obtained by subtracting the conductivity at 14 T from that at zero field. The results are shown in Figure 2c. The $\sigma_{wl}$ term of the weak localization is given by
(3)$$\begin{array}{l} \sigma_{wl}=\frac{e^{2}}{2\pi^{2}\hbar }[3\sqrt{\frac{1}{D\tau_{SO}}+\frac{1}{4D\tau_{i}}}-\sqrt{\frac{1}{4D\tau_{i}}}]+\\ \frac{e^{2}}{2\pi^{2}\hbar }{\left[\frac{eB}{\hbar }\right]}^{1/2}\{f_{3}(\frac{B}{B_{2}})+\frac{1}{2\sqrt{1-\gamma }}[f_{3}(\frac{B}{B_{+}})-f_{3}(\frac{B}{B_{-}})]\}-\\ \frac{e^{2}}{\pi^{2}\hbar }{\left[\frac{eB_{SO}}{3\hbar }\right]}^{1/2}[\frac{1}{\sqrt{1-\gamma }}(\sqrt{t_{-}}-\sqrt{t_{+}})-\sqrt{t}+\sqrt{t+1}] \end{array}$$
with the inelastic scattering time $\tau_{i}$, the spin–orbit scattering time $\tau_{SO}$, the magnetic induction *B*, the equivalent fields $B_{i}=\frac{\hbar }{4eD\tau_{i}}$, $B_{SO}=\frac{\hbar }{4eD\tau_{SO}}$, $\gamma ={\left(\frac{3g\mu_{B}B}{8eDB_{SO}}\right)}^{2}$, $B_{\pm }=B_{i}+(2/3)B_{SO}[1\pm \sqrt{1-\gamma }]$, $B_{2}=B_{i}+(3/4)B_{SO}$, $t=3B_{i}/(4B_{SO})$, and $t_{\pm }=t+(1/2)[1\pm \sqrt{1-\gamma }]$ [23]. The function of *f*_3_ is defined in Reference [24]. At zero field, *B* = 0 in RuSb_2+_, and γ=0, $t_{+}=t+1$, $t_{-}=t$. Therefore, both the second and third terms are zero. Equation (3) becomes $\sigma_{wl}=\frac{e^{2}}{\hbar }[3\sqrt{\frac{1}{D\tau_{SO}}+\frac{1}{4D\tau_{i}}}-\sqrt{\frac{1}{4D\tau_{i}}}]$. $\tau_{i}$ has a temperature dependence as $\tau_{i}=CT^{-p}$, with p≥2 [23]. As RuSb_2+_ is composed of heavy atoms, $\tau_{SO}$ is expected to be very small and significantly influence $\sigma_{wl}$. It is observed that $\tau_{SO}$ decreases by addition of heavy atoms with increased spin–orbit coupling [25]. Considering that the temperature increases the spin–orbit relaxation time, it is assumed that $\tau_{SO}\sim T^{\delta }$ at low temperature. At low temperature, $\tau_{i}>>\tau_{SO}$, so that $\sigma_{wl}\propto T^{-\delta /2}$ is obtained. As shown in Figure 2c, the conductivity contributed by the weak localization is the difference between 14 and 0 T. Below 25 K, the fitting gives $\sigma_{wl}\sim T^{-0.26}$ with δ≈0.52. The low δ value is consistent with the stable spin–orbit interaction with the temperature. As the temperature increases, $\tau_{i}$ becomes compatible with $\tau_{SO}$, and $\tau_{i}$ cannot be ignored. With a low δ value, $\tau_{SO}$ is assumed to be a constant at a narrow temperature range. The major temperature factor in $\sigma_{wl}$ is $\tau_{i}$, and $\sigma_{wl}\propto T$ is obtained. As shown in Figure 2c, it is linear between 25 and 50 K. Furthermore, at *T*_m2_ ~ 312 K, the slope of the resistivity changes from positive to negative, as shown in Figure 2a. It is due to the competing effect of thermally activated and impurity-induced conduction in semiconductors [16], instead of a metallic-insulating transition.

Under the applied magnetic field, the resistivity, measured in either the *ab* or *ac* plane, becomes higher in the whole temperature range. MR is defined as $MR=\frac{R(H)-R(0)}{R(0)}\times 100\%$. MR measured with the current in the *ab* plane and the applied magnetic field out of the *ab* plane is shown in Figure 3a, and that measured with the current in the *ac* plane and the applied magnetic field out of the *ac* plane is shown in Figure 3b, and that measured with the current and the applied magnetic field in the *ab* plane is shown in Figure 3c. Below *T* < 12 K, as shown above, $\rho_{ab}$ has a slow upturn at 0 T. Both the transversal and longitudinal MR (*ab*) are very high. It is consistent with the semimetal nature of RuSb_2+_. In the range of *T*_f_ < *T* < *T*_m1_, both the transversal and the longitudinal MR (*ab*) decrease quickly but are still higher than that in the range of *T*_m1_ < *T* < *T*_m2_. While above *T* > *T*_m2_, MR (*ab*) and MR (*ac*) decrease very fast.

Figure 3a illustrates the transversal MR (*ab*) at different temperatures, 2, 10, 20, 35, 50, 100, 200, and 300 K. At 2 K, the positive MR (*ab*) is as large as 72% at 14 T. The increasing temperature decreases MR (*ab*). At 300 K, it is still 4.5%. Moreover, MR (*ab*) is not saturated at 14 T. MR (*ac*), which is measured on the *ac* plane with the magnetic field out of the plane, is 44% at 2 K, Figure 3b. MR (*ac*) is much smaller than the peer MR (*ab*) at 2 K. Similar to the resistivity, MR is also anisotropic. Analyzing the origin of the MR, the high longitudinal MR (*ab*), about 82% at 2 K, as shown in Figure 3b, is against the Lorentz force effect at low temperature. The Lorentz force effect may contribute to MR above 50 K in RuSb_2+_. Generally, the contribution by the Lorentz force is at zero or first order of *kT*/*E*_f_. It is also consistent with the variation of MR above 50 K. At low temperature, the slow upturning $\rho_{ab}\sim T$ plays an important role on the high MR (*ab*). The MR at low temperature is related to the quantum interference effect, taking the weak localization correction into account. Table 1 lists the MR in other semiconductors with either magnetic or nonmagnetic properties. The transversal MR (*ab*) in RuSb_2+_ is higher than that in other semiconductors.

By investigating the MR, some information about the Fermi surface can be obtained. Kohler’s rule describes the scaling law of the MR with temperature. If the MR measured at different temperatures are scalable with the variable $H/\rho_{0}$, the energy band is a single band, and the Fermi surface is symmetric. The scaling of the RuSb_2+_ single crystal based on Kohler’s rule is shown in Figure 4a. Obviously, MR measured at different temperatures do not fall on the same curve. It indicates that the RuSb_2+_ single crystal does not obey the Kohler’s rule. The discrepancy supports that the RuSb_2+_ single crystal has a multi-carrier transport. For two-band or multiband materials, the MR is described by the empirical equation as $\rho_{xx}=\frac{A+BH^{2}}{C+DH^{2}}$. The MR of the RuSb_2+_ single crystal follows this rule very well, as shown in the inset of Figure 3a. Previously, it has been reported that the Hall resistivity of RuSb_2+_ changes nonlinearly with the magnetic field [19], as shown in Figure 4b. It is consistent with the multiband nature. Furthermore, the Seebeck factor (S) is positive at 300 K, and it decreases with temperature and becomes negative below 60 K, as shown in the inset of Figure 4b [19]. It supports that both positive and negative carriers coexist in RuSb_2+_.

In order to study the nature of the band gap of RuSb_2+_, the UV–vis spectra are measured. Figure 5a shows $\left[F(R)hv^{2}\right]$ versus hv. *F*(*R*) is calculated from the Kubelka–Munk function $F(R)=(1-R^{2})/2R$, with *R* as the measured reflection coefficient and hv as the energy of the incident photon [21]. For a direct allowed transition, the band gap energy is the interception at the low-energy side of $\left[F(R)hv^{2}\right]$ versus hv. The deduced band gap energy is 1.29 eV. An obvious absorption is also observed at 1.29 eV, as pointed out by the arrow in the upper inset of Figure 5a. The direct band gap is confirmed by the Tauc relation, which is described as
(4)$$\alpha hv=K{\left(hv-E_{g}\right)}^{n}$$
with α as the absorption coefficient, K as the system-dependent parameter. While *n* = 1/2, it is direct allowed transition, *n* = 3/2 for direct forbidden transition, and *n* = 2 for indirect allowed transition, and *n* = 3 for indirect forbidden transition [21]. The index *n* is obtained from the logarithmic form of Equation (4),
(5)$$\ln(\alpha hv)=\ln K+n\ln(hv-E_{g})$$
where *n* ≈ 0.6 is derived from the slope of $\ln(\alpha hv)\sim \ln(hv-E_{g})$, as shown in the lower inset of Figure 5a. Therefore, RuSb_2+_ has a direct allowed transition. The deviation from 0.5 is owed to the fractal nature of the density of states due to the disorder in the system [21]. The band gap was reported to be 0.79 eV at 10 K [20]. The difference between the two results is due to the changed energy band structure by the extra Sb in RuSb_2+_ [19].

In order to further evaluate the band structure, the XPS valence band spectra are measured and shown in Figure 5b. The valence band is at 0.39 eV below the Fermi energy. The structures, which are pointed out by the arrows, feature Ru 4*d* electrons [18]. Therefore, the energy band diagram of RuSb_2+_ is proposed and shown in Figure 6. According to the calculation, the *d* electrons of the Ru atom have *t_2g_* and *e_g_* states, and *t_2g_* is lower than *e_g_*. The energy band gap is formed by the separation between the *t_2g_* and the valence band.

## 4. Conclusions

A large unsaturated positive MR is observed in single-crystal RuSb_2+_, especially on the *ab* plane. A robust slow upturning resistivity in the $\rho_{ab}\sim T$ curve is observed. The MR is smaller on the *ac* plane without the slow upturning $\rho_{ac}\sim T$. The MR at low temperature is interpreted as the quantum interference effect, taking both the electronic interaction and weak localization correction into account. RuSb_2+_ has both positive and negative carriers deduced from Hall resistivity and Seebeck coefficient. RuSb_2+_ is a direct band gap semiconductor, with the band gap as 1.29 eV. The valence band lies at 0.39 eV below the Fermi energy. We proposed the schematic energy band diagram of RuSb_2+_ based on the experimental results. Similar robust flat $\rho_{ab}\sim T$ curve and high MR were also observed in WTe_2_, MoTe_2_, NbSb_2_, etc. [10,11,25]. It indicates that RuSb_2+_ may have similar semimetallic properties as in WTe_2_, MoTe_2_, NbSb_2_, etc. However, unlike WTe_2_, RuSb_2+_ is not a topological insulator. In the future, it is worth studying the difference between WTe_2_ and RuSb_2+_.

## Figures and Tables

**Figure 1 materials-13-03159-f001:**
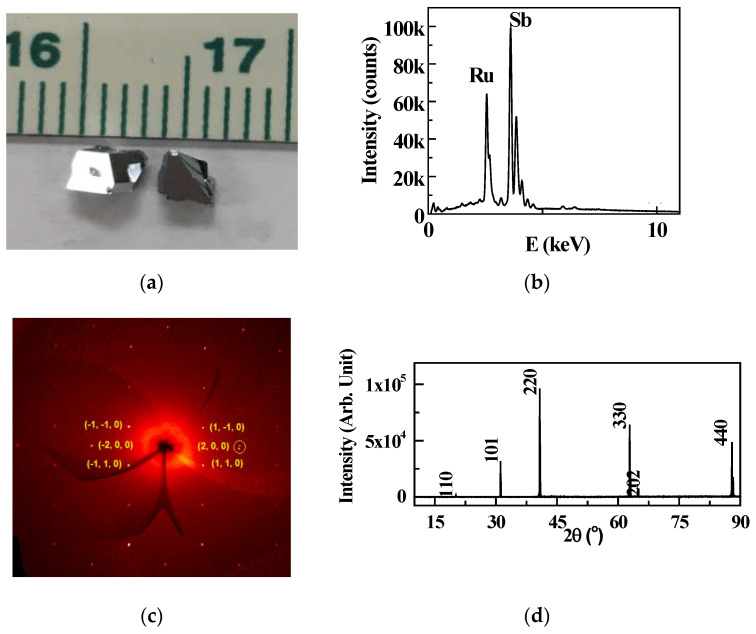
(**a**) Image of a single crystal; (**b**) energy dispersion X-ray (EDX) image; (**c**) XRD procession image on a single crystal, with the number as the (h k 0) index; (**d**) powder XRD patterns measured on the *ac* and *ab* planes of RuSb_2+_ single crystal.

**Figure 2 materials-13-03159-f002:**
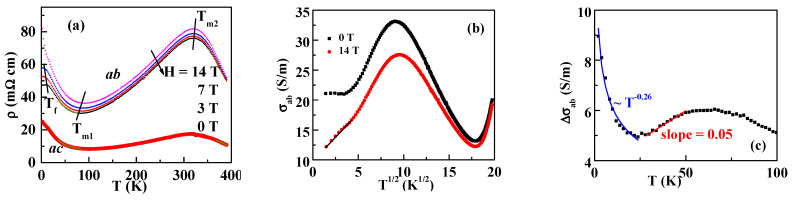
(**a**) Resistivity versus the temperature of RuSb_2+_ on the *ac* plane ($\rho_{ac}$) at 0 T and on the *ab* plane ($\rho_{ab}$) at 0, 3, 7, and 14 T, (**b**) conductivity $\sigma_{ab}$ versus $\sqrt{T}$, and (**c**) conductivity difference between 0 and 14 T. The solid symbols represent the experimental data, and the solid lines are fitting lines as described in the text.

**Figure 3 materials-13-03159-f003:**
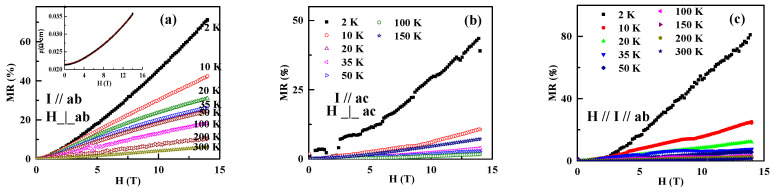
(**a**) The transversal MR measured with the current *I* on the *ab* plane and the magnetic field *H* out of the *ab* plane at 2, 10, 20, 35, 50, 100, 200, and 300 K; (**b**) the longitudinal MR with both *I* and *H* in the same direction on the *ab* plane; and (**c**) the transversal MR measured on the *ac* plane.

**Figure 4 materials-13-03159-f004:**
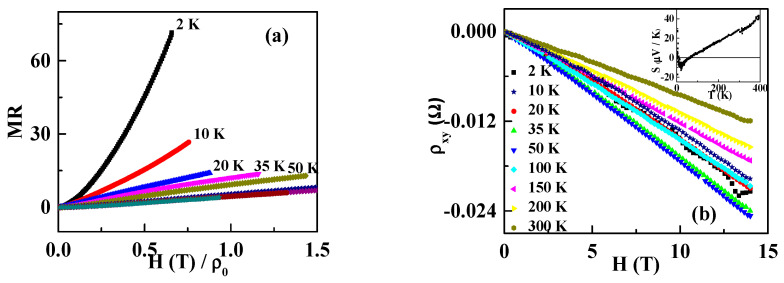
(**a**) The analysis of MR based on the Kohler’s rule; (**b**) the Hall resistivity at different temperatures, and the inset is the variation of the Seebeck coefficient S with temperature.

**Figure 5 materials-13-03159-f005:**
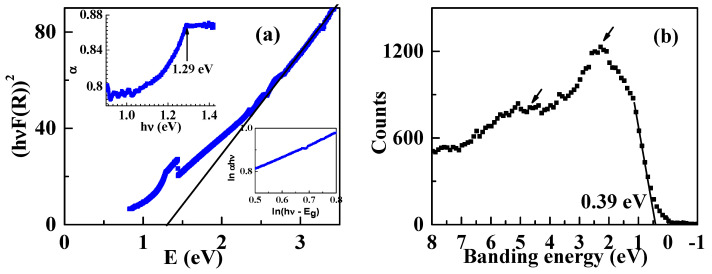
(**a**) ${\left[hvF(R)\right]}^{2}$ versus the energy *E*, and the upper inset is the corresponding absorption curve, and the lower inset is ln(αhv) versus $\ln(hv-E_{g})$; (**b**) the valence band of RuSb_2+_.

**Figure 6 materials-13-03159-f006:**
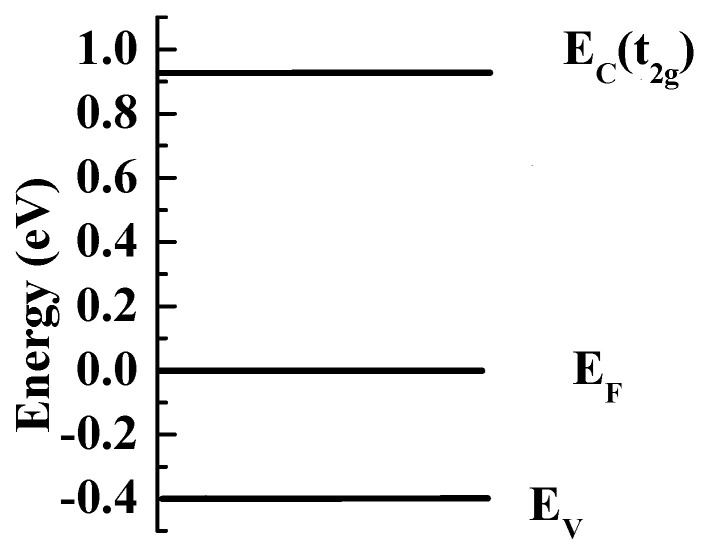
The schematic energy band diagram of RuSb_2+_.

**Table 1 materials-13-03159-t001:** Comparison of the MR observed in RuSb_2_ and other semiconductors.

Compounds	Form	MR @ Magnetic Field and Temperature	Magnetic Properties	Ref.
LuPd_2_Si	polycrystal	21% @ 8 T, 10 K	Magnetic	[26]
Tb_0.5_Lu_0.5_Si_3_	polycrystal	60% @ 12 T, 5 K	Magnetic	[27]
Zn_0.95_Cu_0.05_Cr_2_Se_4_	polycrystal	>80% @ 7 T, 3.2 K	Magnetic	[28]
CdS	pingle-crystal	1% @ 8 T, 1.2 K	Nonmagnetic	[4]
GaAs	film	2% @ 0.9 T, 50 K	Nonmagnetic	[6]
RuSb_2+d_	single crystal	82% @ 14 T, 2 K	Nonmagnetic	Present work

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
