# Peer review of "High Positive MR and Energy Band Structure of RuSb2+"

_materials, 2020, doi:10.3390/ma13143159_

Round 1
Reviewer 1 Report
Dear Editor,
The manuscript “Band Gap and High Positive Magnetoresistance of RuSb” by Liang Zhang, Yun Wang and Hong Chang studies magneto-resistance and other properties of the RuSb semiconductor. The manuscript is poorly prepared and is very hard to read and judge. Author should first fix all misprints, errors and typos. After that the essence of the work can be judged.
Below, please find some of the issues with the manuscript. But there are much more in the text.
- There are some misprint unexpected empty spaces and missed characters in the manuscript. Authors should read it carefully and fix all the problems.
- English language should be also improved. Sometimes it is hard to understand, what authors are saying.
- In the text below Fig. 1 authors mention resistivity in the ac plane, but denote it as .
- Inset in Fig. 2 is not explained in the caption.
- Equations in page 4 are not numerated.
- Lines 96 – 97. Authors introduces characters that are not used in the equation above.
- Second equation in page 4 is incorrect. There should be e^2/h not e^/h^2. Normalized field h and Plank constant are both denoted by the same letter.
- Function f3 in the equation for the localization correction is not introduced.
Etc.
Author Response
We appreciate the reviewer’s comments, which definitely increase the readability of the article.
Comments 1: ‘There are some misprint unexpected empty spaces and missed characters in the manuscript. Authors should read it carefully and fix all the problems.’
Reply: The manuscript is revised to clear those unreadable characters.
Comments 2: ‘English language should be also improved. Sometimes it is hard to understand, what authors are saying.’
Reply: English and the expression in the manuscript is improved to make the article easy to understand.
Comments 3: ‘In the text below Fig. 1 authors mention resistivity in the ac plane, but denote it as .’
Reply: It is corrected.
Comments 4: ‘Inset in Fig. 2 is not explained in the caption.’
Reply: The inset in Fig. 2 is explained in the caption.
Comments 5: ‘Equations in page 4 are not numerated.’
Reply: The Equations in the revised version are numerated.
Comments 6: ‘Lines 96 – 97. Authors introduces characters that are not used in the equation above.’
Reply: Those are corrected.
Comments 7: ‘Second equation in page 4 is incorrect. There should be e^2/h not e^/h^2. Normalized field h and Plank constant are both denoted by the same letter.’
Reply: The error about e^2/h is corrected. In the revised equation, the plank constant, h bar, is used instead of h.
Comments 8: ‘Function f3 in the equation for the localization correction is not introduced.’
Reply: The definition of Function f3 is given by referring a publication on Page 8. Since the definition of f3 is complicated and has little influence on the current analysis, we did not put more definition in the text .
Reviewer 2 Report
high positive magnetoresistance (MR), 78%, is observed at 2 K on the ab plane of the diamagnetic RuSb2+ semiconductor. On the ac plane, MR is 44% at 2 K and about 7% at 300 K. MR at different temperatures do not follow the Kohler’s rule. It suggests that the multiband effect or spin scattering plays a role on the carrier transportation. The nonlinear Hall resistivity and that the Seebeck coefficient changes its sign from positive to negative support that RuSb2+ is a semiconductor with both positive and negative carriers. The high positive MR is due to the quantum interference effect by considering both the interaction and the weak localization correction. Judged from the UV-vis spectra, it is a direct band gap semiconductor with the band gap as 1.29eV. The Valence band being at 0.39 eV below the Fermi energy. The schematic energy band structure is proposed based on experimental results.
The idea behind this is interesting. However, I still have quite a number of concerns in this manuscript. There are times where there are not enough data to support the conclusions of the author. Please see some of the major concerns below.
1. The information for the structures showing in figure 1 is not enough. The authors should give much more information about this. So the readers can get its reproducibility. The labels are not clear need to be modified and the numbering in the image (1(b)) are not clear.
- The authors should give much more information about the novelty of this paper, especially the effect of using this RuSb2+ with both positive and negative carriers, which applications can be used this semiconductor and utilized it is MR results?
- More references need to be included in the introduction part to understand the applications of using silicon semiconductor that has also positive and negative carriers such as,
a.Design of fiber-integrated tunable thermo-optic C-band filter based on coated silicon slab"J. of the European Optical Society-Rapid Publications, 13, 2017 (32)
b.Improving Raman spectra of pure silicon using super-resolved methodJ. of Optics, 21(7), 2019 (075801 – 6 pages)
- To add a table that comparison between MR in semiconductor materials that have both positive and negative carriers
- Much more discussion about the results should be given in this paper, especially the author needs to provide enough physicals mechanism analysis about the results.
Author Response
We appreciate the reviewer’s comments, which definitely increase the readability of the article.
Comments 1: ‘The information for the structures showing in figure 1 is not enough. The authors should give much more information about this. So the readers can get its reproducibility. The labels are not clear need to be modified and the numbering in the image (1(b)) are not clear.’
Reply: Considering that it has little relation with the current research, the structure in Figure 1 is deleted. The number in Figure 1 (b) is explained in the caption in the revised manuscript.
Comments 2: ‘The authors should give much more information about the novelty of this paper, especially the effect of using this RuSb2+ with both positive and negative carriers, which applications can be used this semiconductor and utilized it is MR results?’
Reply: More information about the novelty is given on page 3 in the introduction.
‘Even though high MR is observed in devices combining a semiconductor and a magnetic material, such as spin valve transistors [22], magnetic tunnel transistors [23] and stray-field-induced MR devices [24], it is seldom observed in a single crystal. MR in nonmagnetic single crystals is closely related to the changed electronic density of state or the energy band structure by the applied field. MR in nonmagnetic single crystals gives a clue of other interesting physical phenomena.’
Comments 3: ‘More references need to be included in the introduction part to understand the applications of using silicon semiconductor that has also positive and negative carriers such as,’
Reply: More references are added in the introduction and the other parts of the text.
Comments 4: ‘To add a table that comparison between MR in semiconductor materials that have both positive and negative carriers’
Reply: One table is added to compare MR with other semiconductor materials.
Comments 5: ‘Much more discussion about the results should be given in this paper, especially the author needs to provide enough physicals mechanism analysis about the results’.
Reply: The physical mechanism is analyzed based on the quantum interference effect by taking both the weak localization correction into consideration. The text is reorganized to make it clearer.
Reviewer 3 Report
The manuscript entitled Band Gap and High Positive Magnetoresistance of RuSb2+ reports on the transport properties of high flux-fabricated crystals of RuSb2+, meaning it is not stoichiometric but Sb rich. The authors also determined the bandgap from optical absorption (UV-Vis) spectroscopy besides proposing a simple energy level band diagram. While such studies are timely given the fact a significant growth in Quantum Materials (Weyl semimetals, topological insulators etc.) research, BUT this paper is extremely poorly presented ad needs major rework. It is extremely hard to follow and understand their scientific explanation and the presentation appears scattered and as the first draft of a chapter. Also, there are numerous mistakes throughout the manuscript as far as notations and symbols are concerned. The quality of figures is also not high. Additionally, the title can be revised and one must expand the acronyms the first time authors introduce in the text. Given several drawbacks in this manuscript, it is far from consideration for or re-consideration for publication in this high-impact journal of Materials. Therefore the manuscript in its current state must be rejected.
Author Response
Comments: ‘It is extremely hard to follow and understand their scientific explanation and the presentation appears scattered and as the first draft of a chapter. Also, there are numerous mistakes throughout the manuscript as far as notations and symbols are concerned. The quality of figures is also not high. Additionally, the title can be revised and one must expand the acronyms the first time authors introduce in the text. ’
Reply: The article is rewritten to make the scientific explanation clear.
The article is revised to correct the unreadable notations and symbols.
Some adjustments are made on the figures.
The title is changed to ‘High Positive MR and Energy Band Structure of RuSb2+’.
The acronyms are defined at the first time in the revised version.
Reviewer 4 Report
The authors performed an excellent study. The manuscript is prepared well and has the potential to have a significant impact on the area.
Similar results were shown in recent works by other researchers, but the key strength of this is manuscript is preparing single crystal in the lab and continue to the experiment. This adds a lot of value to the experimental data.
The reviewer would like to know what are the unique findings this manuscript claims. Also, what is the next step the authors would like to take on this area?
There are some typo in the manuscript needs to be corrected.
- Line 96, 97, 101 See/Swl needs to be in symbols
- Line 160 missing r
- Line 216 missing ab
Author Response
We sincerely appreciate the reviewer’s comments. We will also benefit from these comments in the future.
Comments 1: ‘The reviewer would like to know what are the unique findings this manuscript claims. Also, what is the next step the authors would like to take on this area?’
Reply: Some are added on page 3 in the introduction.
‘Even though high MR is observed in devices combining a semiconductor and a magnetic material, such as spin valve transistors [22], magnetic tunnel transistors [23] and stray-field-induced MR devices [24], it is seldom observed in a single crystal. MR in nonmagnetic single crystals is closely related to the changed density of state or the energy band structure by the applied field. MR in nonmagnetic gives a probable clue of other interesting physical phenomenon. ’
Some are added on page 15 in the conclusion.
‘It indicates that RuSb2 below 12 K may have similar semimetallic properties as in WTe2, MoTe2, NbSb2, etc. However, unlike WTe2, RuSb2+ is not a topological insulator. In the future, it would be worth to study what results in the difference between WTe2 and RuSb2+.’
Comments 2: ‘There are some typo in the manuscript needs to be corrected.’
Reply: Those typo are corrected in the revised manuscript.
Round 2
Reviewer 1 Report
Dear Editor,
I reviewed the revised manuscript “Band Gap and High Positive Magnetoresistance of RuSb” by Liang Zhang, Yun Wang and Hong Chang. While authors make some improvements on the presentation style the manuscripts is still very far from being well prepared for publication. Language is still poor, equations are still wrong and not well described, figures can be also improved. Additionally, the data presented is questionable and contradictive. So, I strongly suggest not to publish the manuscript in the present form.
See attached file for more details.

Author Response
We would like to express our appreciation to the referee’s comments. Those help us improve the manuscript’s quality and avoid some scientific errors.
Comments 1: ‘While I’m not a native English speaker I fill that there are many issues with language. For example, in the abstract authors write: “The quantum interference effect lies behind high positive MR taking both the weak localization correction into consideration”. What does the word “both” refer to?
A couple of sentences later: “The Valence band being at 0.39 eV below the 14 Fermi energy.” Why the word valence starts with capital letter. The grammar is also incorrect in the sentence. What is the verb here?
Introduction: “As semiconductors are noted to have many applications [1,2], MR is also observed in heavily doped n-type nonmagnetic 21 semiconductors, such as Ge, Si, GaAs, CdS etc.” How does the first part of the sentence relate to the second part.
Introduction: “In organic semiconducting MR, …”. What is the “semiconducting
magnetoresistance”?
Introduction: “The effect of the magnetic doping on the thermoelectric doping …”. What is “thermoelectric doping”? Etc.
Finally, I suggest to authors that they ask some native English speaker to check the manuscript.’
Reply: We carefully checked English in the newly revised manuscript. The errors are corrected in the introduction on page 1.
Comments 2: ‘The statement “MR is generally observed in spin-polarized materials when 19 the spin-up and spin-down carriers experience different conduction routes.” looks strange. The MR is observed in normal metals which do not have electron spin polarization.’
Reply: The statement is rewritten in the introduction on line 8-10 of page 1 as ‘In spin-polarized materials, MR is observed as the spin-up and spin-down carriers experience different conduction routes. Semiconductors are an important series of materials with many interesting properties [1,2].’.
Comments 3: ‘The inset in Fig. 2 is still not explained in a proper manner and it seems that the inset contradicts to the main figure. Does it show a derivative as a function of and ? Is the derivative zero in the whole region below the lower line in the inset (below 15 K)? Or the derivative is zero only at this line? What is the upper line in the inset? Does it split the regions with positive and negative ?
In the main figure the curve for 14 T does not show any extremum below 50 K (it just decay monotonically). This means that there is no region or point at which the derivative is zero. In contrast, in the inset there is a region of temperatures (below 10 K) for where.
The symbols and axes tics in this inset are very small (may be this is a reason for
misunderstanding of what is show). Since in Fig. 2 there are just three panels, authors can make the inset bigger and put it as a 4-th panel in Fig.2.’
Reply: About the inset of Fig. 2, we initially intended to make an MR phase diagram. However, considering that it did not improve the quality of the manuscript, we deleted the inset of Fig. 2 in the revised version. Without it, the whole context is even clearer.
Comments 4: ‘What are the solid lines in Fig. 2(b and c)? Is this just a guide for eye or some simulations?’
Reply: The solid lines are the fitting curves. In the main text and the caption of Fig. 2 on page 3 and 4, it was clarified. Furthermore, Fig. 2(b and c), on page 3 and 4, are redrawn to make them fit the text.
Comments 5: ‘Fonts change all the time. Some symbols are small some are big. For example, the notation of the derivative in the caption of Fig. 2 is big comparing to other text in the caption. Above line 124 the font is big while staring from line 125 it is small.’
Reply: Fonts are in the same size in the revised version.
Comments 6: ‘Some notations are still not explained. For example, the function in Eq. (2) is not introduced. Even magnetic field is not introduced in the text. Relations between magnetic field and reduced magnetic filed is not introduced.’
Reply: In the revised manuscript, those notations are explained in the text on page 4, including the function g3 on line 92, the magnetic field and the relation between magnetic field and reduced magnetic filed on page 4 and 5.
Comments 7: ‘Below Eq. (2) is explained as reduced Plank constant while in fact it is a normalized magnetic field (see Ref. [25] where Eq. (2) is taken from).’
Reply: The Plank constant and the normalized magnetic field are separated as h bar and h in the revised version on page 4.
Comments 8: ‘In Eq. (3) is indeed should be the Plank constant. This introduces a big confusion.’
Reply: The Plank constant and the normalized magnetic field are separated as h bar and h in the revised version on page 4.
Comments 9: ‘In Eq. (3) fields , are not defined. The expression Eq. (3) itself is different comparing to the expression in Ref. [25] where it is taken from. According to Ref. [25] there should be a factor in front of the last term. This factor is absent in the current manuscript without any explanation. Also, below Eq. (3) authors refer to Ref. [22] which is also wrong. Ref. [22] does not discuss the weak localization at all.’
Reply: In the revised version on page 5, fields in Eq. (3) are defined. Eq. (3) is corrected according to Ref. [23]. The reference is corrected.
Comments 10: ‘In Fig. 2c the difference between conductivity at 14 T and at 0 T change sign. It is not clear since the resistivity at 14 T is always higher that the resistivity at 0 T. They never intersect. What is the expression plotted in Fig. 2c?’
Reply: Fig. 2c was wrong in the previous version. In the revised version, it was corrected on page 4.
Comments 11: ‘After Eq. (3) authors state: “As the temperature is high enough. The slope of the conductivity to the temperature is …, which is negative, i.e. positive for the resistivity”. The sentence is absolutely unclear from grammar point of view and from physical one. At high temperature only governs the temperature dependence of the conductivity. At that the temperature dependence is not discussed in the text. The expression for the conductivity derivative with respect to temperature looks incorrect at all. Where is in it? Moreover the statement completely contradicts the curve shown in Fig. 2c. At high temperature the conductivity increases not decreases.’
Reply: We carefully think over this part. The spin-orbit relaxation time is the key. However, the research on the spin-relaxation theory is not complete. Supposed that the spin-orbit relaxation time is proportional to T^delta, the experimental data gives a very low delta. It is also consistent with spin-orbit’s slow reaction to the temperature. The discussions are on Page 5.
Comments 12: ‘The rest of the manuscript also has many issues.’
Reply: We have carefully revised the manuscript to make it clear and free of English error.

Reviewer 2 Report
The new version can be published
Author Response
We made some change in the revised manuscript, and we also corrected some English error.
Reviewer 3 Report
The revised manuscript addressed the points raised with additional changes in the title and abstract. As a result of these revisions, the manuscript reads very well now and it is acceptable for its re-consideration after minor but important editorial deficiencies.
Author Response
We made some adjustment in the introduction and corrected some English error in the revised manuscript.
Round 3
Reviewer 1 Report
Dear Editor,
I reviewed the second revision of the manuscript “Band Gap and High Positive Magnetoresistance of RuSb” by Liang Zhang, Yun Wang and Hong Chang. Authors made improvements to the manuscript but there are still some negligences and data inconsistencies. Therefore, I can not recommend the manuscript for publication in the present form. Below please, find my comments.
- Caption of Fig. 1. “PrOcession image” instead of “precession image”.
- Quantities t, t_{\pm} in Eq. (3) are not introduced.
- After Eq. (3) authors discuss the temperature dependence of the weak localization correction to conductivity. They claim that it behaves as T^{-\delta/2}. This statement is in fact does not look proven. The first term in the Eq. (3) indeed scales as T^{-\delta/2}. However, the last term should scale in a different way. Sqrt(B) chancel for the last term and its behavior is defined by quantities t and t_{\pm} which are not introduced in the text. If they depend on temperature the whole expression has different temperature variation.
- It also looks as a contradiction that at 14 T the conductivity scales as sqrt(T) but difference between conductivities at 14 T and 0 T behaves as T^{-0.26}. The conductivity at 0 T is flat so the difference between the conductivities should scales as C-\sqrt(T) instead of T^{-0.26}. Can authors explain this?
- Can authors clarify what they mean by “As analyzing the origin of the MR, the Lorentz-force is ruled out as it is at zero or first order of kT/Ef, which is not the case in RuSb2+.” Does it mean that Lorentz force induced MR linearly depends on temperature? From Fig. 3 I see that the MR(T) indeed has a very big linear region. Why authors disregard the Lorentz force as a source of the MR effect?
6. That would be useful for reader if authors provide information on how the magnetic field is oriented in experiments shown in Fig. 2. Then this data can be compared to data in Fig. 3. According to Fig. 2c the weak localization correction behaves in a non-monotonic way. So, I would expect that MR also behaves in non-monotonic way with temperature. But in Fig. 3a, MR monotonically decrease with temperature. Can authors clarify this.
Author Response
We would like to appreciate the reviewer’s sharp comments. Those are very helpful at improving the quality of the manuscript.
Comment #1: Caption of Fig. 1. “PrOcession image” instead of “precession image”.
Reply: It is corrected in the revised manuscript.
Comment #2: Quantities t, t_{\pm} in Eq. (3) are not introduced.
Reply: Quantities t, t_{\pm} in Eq. (3) are introduced on Page 5.
Comment #3: After Eq. (3) authors discuss the temperature dependence of the weak localization correction to conductivity. They claim that it behaves as T^{-\delta/2}. This statement is in fact does not look proven. The first term in the Eq. (3) indeed scales as T^{-\delta/2}. However, the last term should scale in a different way. Sqrt(B) chancel for the last term and its behavior is defined by quantities t and t_{\pm} which are not introduced in the text. If they depend on temperature the whole expression has different temperature variation.
Reply: In the revised version, the quantities of t and t_{\pm} are defined on page 5, and Eq. (3) is rewritten to make it clear. As B = 0, the last term is also zero. In the text on page 5, it is written as ‘At zero field, B = 0 in RuSb2+, and , , . Therefore, both the second and the third terms are zero.’
Comment #4: It also looks as a contradiction that at 14 T the conductivity scales as sqrt(T) but the difference between conductivities at 14 T and 0 T behaves as T^{-0.26}. The conductivity at 0 T is flat so the difference between the conductivities should scales as C-\sqrt(T) instead of T^{-0.26}. Can authors explain this?
Reply: It is a contradiction we did not notice. It was not accurate to describe the conductivity as a constant. The conductivity at 0 T is not a fully constant, but slightly changes. As a result, the difference between conductivities at 14 T and 0 T behaves as T^{-0.26}, which is different from sqrt(T). The description about the conductivity at 0 T has been changed on page 3 and the other part of the text.
‘r_{\ab} has a very slow upturn. In fact, it is almost flat at 0 T.’
Comment #5: Can authors clarify what they mean by “As analyzing the origin of the MR, the Lorentz-force is ruled out as it is at zero or first order of kT/Ef, which is not the case in RuSb2+.” Does it mean that Lorentz force induced MR linearly depends on temperature? From Fig. 3 I see that the MR(T) indeed has a very big linear region. Why authors disregard the Lorentz force as a source of the MR effect?
Reply: The Lorentz-force probably contributes to MR at temperatures higher than 35 K. It is rewritten on page 6,
‘As analyzing the origin of the MR, the high longitudinal MR(ab), about 82% at 2 K as shown in Fig. 3 (b), is against the Lorentz-force effect at low temperature. It may contribute to MR above 35 K in RuSb2+. Generally, the contribution by the Lorentz-force is at zero or first order of kT/Ef. It is also consistent with the variation of MR above 35 K. ’
Comment #6: That would be useful for reader if authors provide information on how the magnetic field is oriented in experiments shown in Fig. 2. Then this data can be compared to data in Fig. 3. According to Fig. 2c the weak localization correction behaves in a non-monotonic way. So, I would expect that MR also behaves in non-monotonic way with temperature. But in Fig. 3a, MR monotonically decrease with temperature. Can authors clarify this.
Reply: How the magnetic field is oriented in experiments in Fig. 2 is described on Page 3 in the revised manuscript. ‘Fig. 2 (a) shows the temperature dependence of the resistivity measured on the ab and ac planes with the magnetic field out of the plane’
About the difference between MR obtained from R~T curve and MR measured at different temperatures, it is due to the different measuring process. As measuring MR, after the temperature is stable at each point, and the magnetic field changes very slowly to and fro between -14 T and 14 T. It allows the carriers to be activated and to precede under the magnetic field. On the other hand, R ~ T is measured at the field of 14 T with the temperature continuously rising. In this case, some of the carriers are stiff and do not participate in preceding.
